# Fully automatic transfer and measurement system for structural superlubric materials

Li Chen [1,2], Cong Lin[3], Diwei Shi[1,2], Xuanyu Huang [2,4], Quanshui Zheng [1,2,5,6], Jinhui Nie [5] ✉ & Ming Ma [2,4,5] ✉

Structural superlubricity, a state of nearly zero friction and no wear between two contact surfaces under relative sliding, holds immense potential for research and application prospects in micro-electro-mechanical systems devices, mechanical engineering, and energy resources. A critical step towards the practical application of structural superlubricity is the mass transfer and high throughput performance evaluation. Limited by the yield rate of material preparation, existing automated systems, such as roll printing or massive stamping, are inadequate for this task. In this paper, a machine learning-assisted system is proposed to realize fully automated selective transfer and tribological performance measurement for structural superlubricity materials. Specifically, the system has a judgment accuracy of over 98% for the selection of micro-scale graphite flakes with structural superlubricity properties and complete the 100 graphite flakes assembly array to form various pre-designed patterns within 100 mins, which is 15 times faster than manual operation. Besides, the system is capable of automatically measuring the tribological performance of over 100 selected flakes on $Si_3N_4$, delivering statistical results for new interface which is beyond the reach of traditional methods. With its high accuracy, efficiency, and robustness, this machine learning-assisted system promotes the fundamental research and practical application of structural superlubricity.

Structural superlubricity (SSL), a state of ultralow friction and no wear between solid surfaces[1,2], is a disruptive technology for friction and wear reduction[3]. It has dramatic application prospects in MEMS[2], such as micro-generators[4], micro-oscillators[5], and others[1,6,7]. At present, graphite flake based on highly oriented pyrolytic graphite (HOPG) is the dominant material for SSL and has realized superlubricity experimentally on various layered heterojunctions[2,8,9] and 3D materials[10,11] thanks to its ultrahigh in-plane strength, weak interlayer interaction, and single crystalline surface[1,12,13]. However, due to the limitation caused by the size of the single crystalline grains of HOPG, after the

fabrication process based on ion etching[2], the size and yield rate of graphite flake with SSL are limited to about 10 μm and 60%, respectively[13-15]. Practical applications such as micro-generator for driving a microelectronic device with a typical power of 1 μW[16,17], usually require the scale of SSL reaching 300 μm[4], which amounts to a requirement of assembling 900 graphite flakes, where the selective transfer of individual flake is the key step.

Besides the demand for efficient transfer, the other requirement is the automatic measurement of the frictional properties for layered heterojunctions. Nowadays, with the boost of 2D materials, there have

[1]Department of Engineering Mechanics, Tsinghua University, Beijing 100084, China. [2]Center for Nano and Micro Mechanics, Tsinghua University, Beijing 100084, China. [3]Department of Computer Science and Engineering, University of California, San Diego, CA 92093, USA. [4]Department of Mechanical Engineering, Tsinghua University, Beijing 100084, China. [5]Research Institute of Tsinghua University in Shenzhen, Shenzhen 518057, China. [6]Institute of Materials Research, Shenzhen International Graduate School, Tsinghua University, Shenzhen 518055, P. R. China. ✉e-mail: niejh@Tsinghua-sz.org; maming16@tsinghua.edu.cn

been thousands of layered heterojunctions that can be fabricated[18]. To find out the system with optimal SSL property among such a large number of candidate systems by experiment is impossible for manual operation. Therefore, automatic, efficient, and reliable measurement techniques are also required.

Auto-transfer methods have been developed recently for promoting the transfer efficiency of microscale materials, such as mass transfer technology for micro-LED[19–23], automatic transfer technology for 2D materials[24–28], and so on, which boost the development of micro-processing and assembly. However, all these methods are difficult to perform micronewton force-resolved manipulation where appropriate probes are required. Specifically, the mass transfer technology represented by micro-LED transfer often relies on PDMS stamping or laser transfer printing[19,20], which requires high sample uniformity and yield rate, making it difficult to apply on the condition of selective transfer or non-homogeneous materials[29–31]. The robotic transfer technology, represented by the automatic transfer of 2D materials, utilized the robot arm for picking the pre-alignment van der Waals solid[25], resulting in difficulty in operating flexibly to evaluate the tribological and mechanical performance of micron materials or devices. Therefore, probe-based selective transfer and auto-measurement techniques for micron material are necessary for the fundamental research and practical application of SSL.

Here, we build a fully automated system using high-precision displacement stages and force sensors. By combining deep learning and image recognition techniques, rapid selection, transfer, and tribological property measurements of SSL materials are achieved. The final selection accuracy exceeds 98%, and the transfer success rate exceeds 94%. The system can perform efficient transfer and measurement on graphite flakes with different sizes (4–10 μm) and different covers (Au, Pt, Al, Ag, $SiO_2$, $Si_3N_4$), indicating the robustness of the system. To demonstrate the system's tribological measurement

ability, the comprehensive friction performance tests for more than 100 selected flakes were conducted automatically after being transferred on a $Si_3N_4$ substrate by the system. These insights will be instrumental in furthering the understanding of SSL materials and expediting their application in various fields.

## Results

### Architecture and functions for the system

The schematic design for the core components of the proposed automatic system is shown in Fig. 1a, which mainly includes two precision force sensors, an *XYZ* precision stage for operation, such as pushing, picking, and placing, an *XYZ* translation stage for large range material transfer, a manual telescopic axis, and a microscope. all those components are connected to the computer except for the manual telescopic axis. The degrees of freedom of the two stages are all controlled by the designed software. To avoid particulate pollution either from the environment or human beings, the entire system is placed in a clean room of class 1000 (at a relative humidity of 55 ± 5% and temperature of 23 ± 2 °C). The fully automatic system also includes accessories such as a vibration isolation table, a high voltage source, controllers, a computer, and a charge-coupled device (CCD) (Fig. 1c). The system stage can accommodate the target substrates with different sizes of 10 × 10 mm, 2-inch and 4-inch. The stage travel range covers the effective area of 4-inch samples with a displacement resolution of 10 nm, which is compatible with existing micromachining systems. Besides, a user-friendly graphical user interface (GUI) has been designed for convenient utilization, where the detailed functions are presented in Supplementary Note 1 and Supplementary Fig. 1 and source codes are provided in Supplementary Software 1. Figure 1d shows a photograph of the core components, where the system transfers a Pt-coated graphite flake onto a 4-inch silicon wafer substrate. The detailed process for generating graphite flakes is described

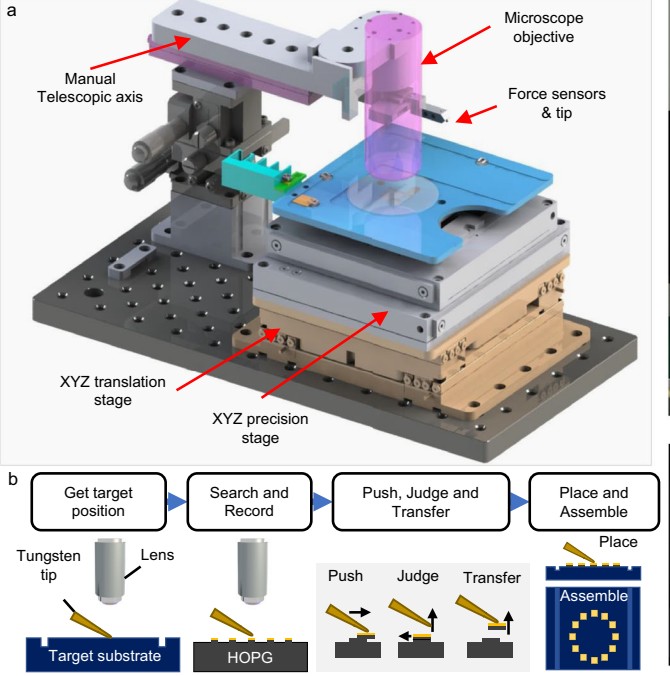

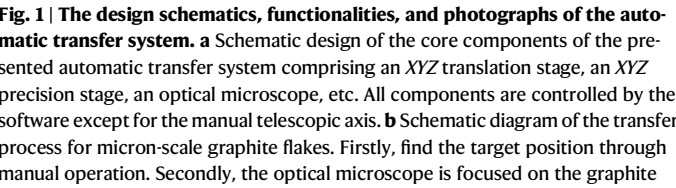

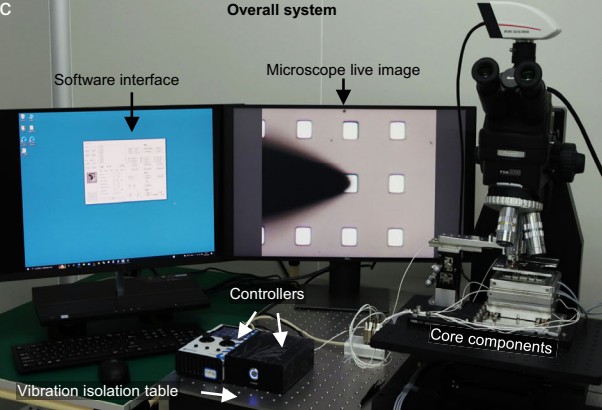

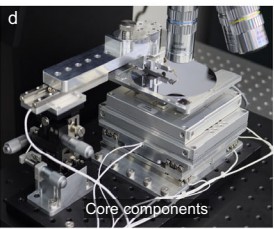

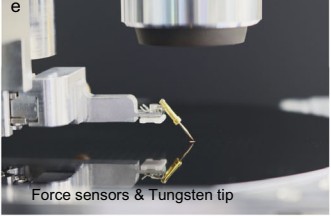

**Fig. 1 | The design schematics, functionalities, and photographs of the automatic transfer system. a** Schematic design of the core components of the presented automatic transfer system comprising an *XYZ* translation stage, an *XYZ* precision stage, an optical microscope, etc. All components are controlled by the software except for the manual telescopic axis. **b** Schematic diagram of the transfer process for micron-scale graphite flakes. Firstly, find the target position through manual operation. Secondly, the optical microscope is focused on the graphite

flakes and the system will automatically find the tip and record the target flakes. Then, the graphite flake is pushed and picked up by the translation stages and probe. At the same time, the system judges whether the graphite flake is an SSL material and whether the operations are successful. Finally, the flakes would be transferred to the target position. **c**–**e** Photographs of the overall system (**c**), close-up of core components (**d**), and force sensors and tungsten tip (**e**).

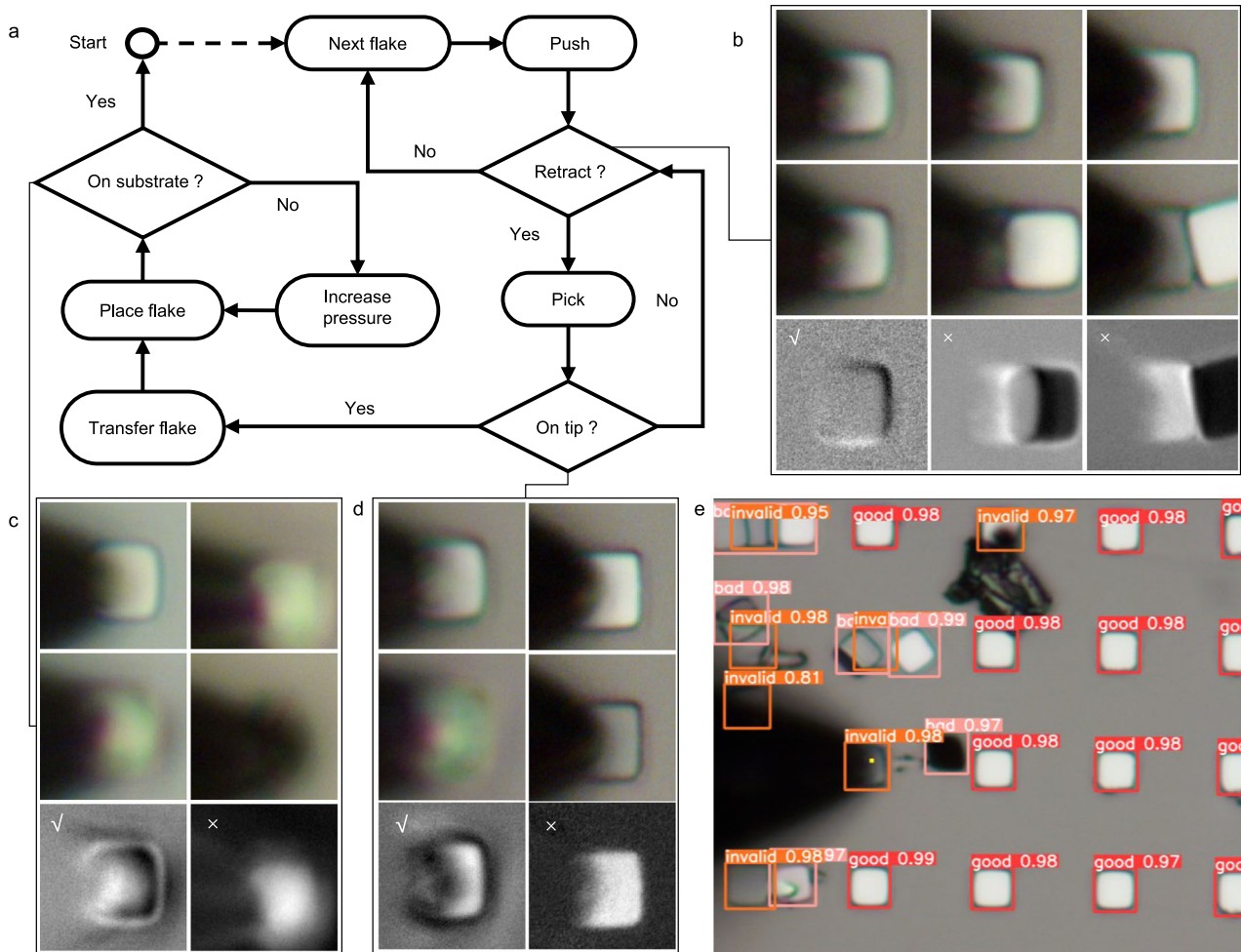

**Fig. 2 | Schematic of a typical SSL flake transfer and examples for several key decisions in the automated transfer process. a** Flow diagram of the automatic transfer system. **b–d** Photographs and processing results for typical sets of critical step decisions, where the first row shows the samples before the operation, the second row presents the images after the operation, and the third row is the difference between the above images, which is also the input to the model: **b** SSL judgments (√ means SSL graphite flakes), (**c**) placement operation judgment (√ means place successfully), (**d**) picking operation judgment (√ means pick up successfully). **e** The result of localizing the tip and the graphite flakes, where the rectangles labeled with "good" denote the target flakes, "invalid" and "bad" denote the contaminated or broken flakes and the yellow point denotes the tip.

in Supplementary Fig. 2 and the Methods section. Figure 1e shows a detailed demonstration of the placement of a graphite flake onto a silicon wafer with a tungsten probe connected to a normal force sensor of 1 μN resolution and a lateral force sensor of 0.2 μN resolution.

The diagrams of the automated transfer workflow are presented in Fig. 1b. First, the system records the position of the target substrate, and calculates the coordinates of the target position with a set pattern (Supplementary Movie 1). Then, after the transfer parameters setting (Supplementary Movie 2), the force sensors, translation stages, and machine learning algorithms work jointly to search and record the nearest graphite flake, push the target flake and determine whether it has SSL property. If the graphite flake shows SSL, the system will pick up the flake utilizing the translation stages and the tungsten probe, translate it to the target position and place the flake onto the substrate. Throughout the transfer, deep learning algorithms are used to monitor the process and determine whether those operations are successful. The entire operation and transfer process are shown in Supplementary Movie 3.

**Deep learning-based localization, selection, and operation**
The main difficulties in the automated transfer or measurement of SSL materials are the positioning and selection of superlubric graphite flakes and the judgments of the operation results. For example, due to

the interference of contaminants and non-SSL flakes (Fig. 2e), conventional image processing methods are unable to accurately locate the tip and graphite flakes nor determine the results of the operations on the graphite flake. To deal with the above difficulties, each determination step in our designed process, with the schematic shown in Fig. 2a, is completed by a custom-trained convolutional neural network (CNN)[32]. Specifically, the system first records the locations of the tungsten tip and all the flakes within the scope (Fig. 2e) through a well-trained network of YOLO-V5[33]. Then, it moves to the nearest flake along the pre-defined direction. Afterward, the system determines whether the graphite flake is superlubric, that is, it will return to its initial position after being pushed away (Fig. 2b). This spontaneous return motion, namely the self-retraction motion[3], suggests the super-lubricity property of the flake. Based on the above characteristic, the model compares the states of the graphite flake before and after the pushing operation and returns the decision. If the flake shows SSL, the system will pick it up by the probe tip through the van der Waals force, otherwise, the system will call the above positioning algorithm and move the tip to the next flake.

After the picking operation, the difference between the graphite flake images before and after the operation will be input to the judgment model to determine whether the flake has adhered to the tip, where Fig. 2d shows the sample input images. If failed but the graphite

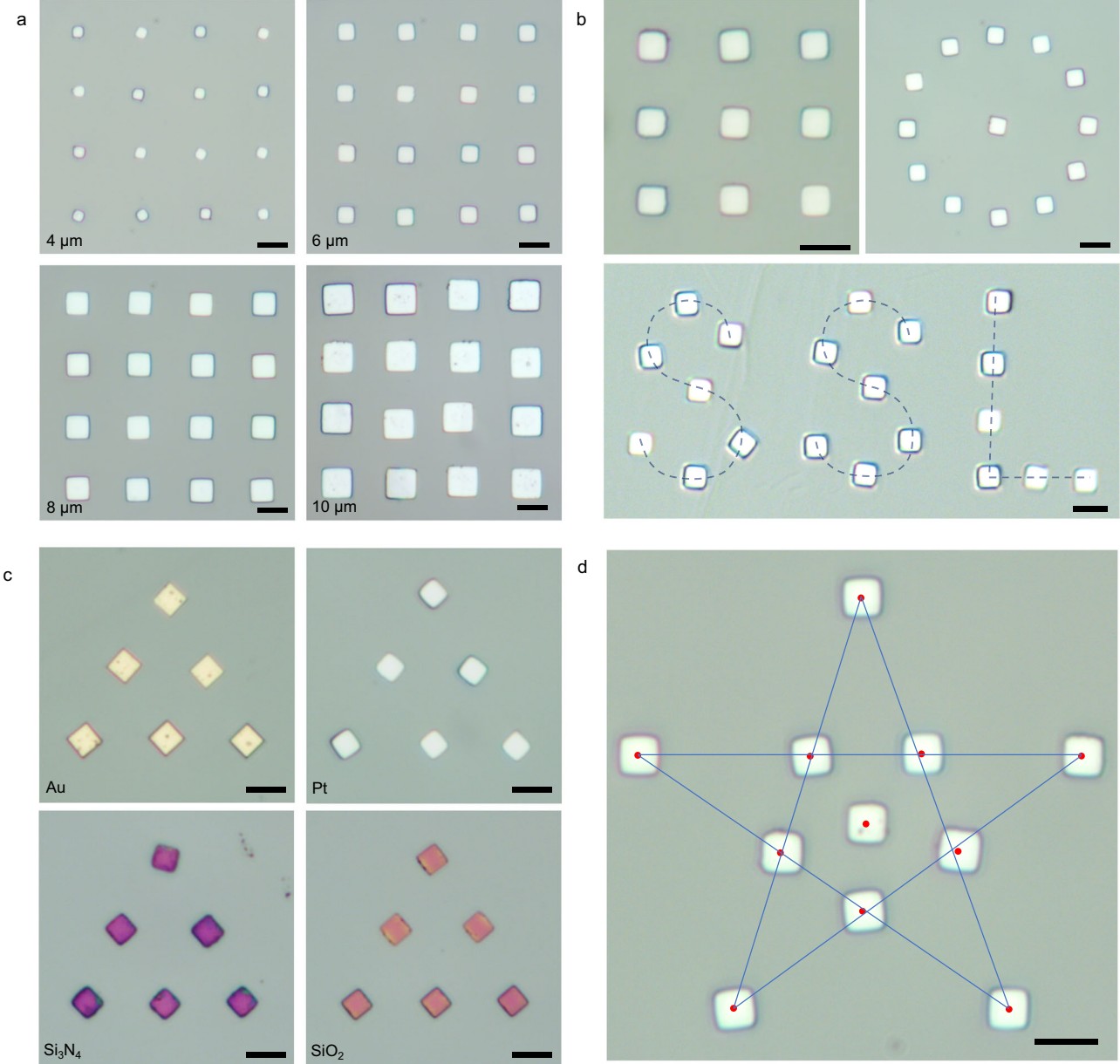

**Fig. 3 | Transfer results for SSL graphite flakes. a** Arrays of graphite flakes of different sizes after transfer. **b** Transfer and arrange flakes into matrices and irregular arrays. **c** Identification and transfer of graphite flakes covered with different materials, demonstrating identification robustness. **d** The shape of the pentagram lined up after the transfer, showing graphite flakes placed within ±0.5 μm of each other. The red dots in (**d**) indicate the center of the graphite flakes and their diameters are all 1 μm. The scale bars in the figures are all 10 μm.

flake returns to its initial position, the system will pick it up again. Otherwise, the system will find the next flake. Once the probe picks a graphite flake, the system transfers the graphite flake to the set target position and judges the placement operation through the same judgment model by predicting whether the flake is on the substrate, with the input image for the examples shown in Fig. 2c. The influence factors of the performance of neural networks are analyzed in Supplementary Note 2 and Supplementary Fig. 8. Besides, to avoid placement failure, the system ensures successful placement by multiple placement operations and increasing the positive pressure during placement.

**Results for automatic transfer**

To elaborate on the transfer capability of the automatic system, the following validation experiments were carried out. First, the system transferred graphite flakes in different sizes with various covers and

arranged them into a given array. Considering that the transfer of graphite flakes mainly relies on the van der Waals force between the tip and the cover materials, the correlations between the normal load for transfer and the size of graphite flake were investigated as shown in Supplementary Fig. 3. Based on the above results, different transfer parameters are designed for flakes in different sizes and various covers, and the experimental results (Fig. 3a, c) demonstrate the robustness of the system.

Then, to meet the requirements of flexibility and accuracy of the transfer pattern and position in practical applications, the functions for a variety of array transfers and custom target transfers have been developed in the system. Figure 3b shows the system automatically transferring graphite flakes into the built-in patterns, such as rectangular, and circular arrays, and the custom set "SSL" glyphs. Meanwhile, in Fig. 3d, the system automatically transferred SSL graphite flakes into pre-set pentagram arrays, with the red dot being the center of the

graphite flake, whose diameter is 1 μm, illustrating that the transfer system achieves an optical limit of ±0.5 μm transfer accuracy. The influence analysis of flake size and tip are presented in Supplementary Note 3 and Supplementary Fig. 9 and the entire transfer operations for assembling the pentagram array are shown in Supplementary Movie 4.

### Automated tribological measurement for SSL materials

To validate the automated tribological measurement ability of the system, a series tests for over 100 graphite flakes with varying sizes were conducted. These flakes were automatically transferred to a flat $Si_3N_4$ surface, with the morphology represented by atomic force microscope (AFM, PARK NX20) shown in Fig. 4a and corresponding mean roughness (Ra) of 0.73 nm. To determine the stable friction force and avoid the interference of factors such as interfacial inclusion, the system automatically located the transferred flake, applied a normal load of 200 μN through a tungsten probe and triggered 200 cycles running-in process[34] on the $Si_3N_4$ substrate, with an amplitude of 5 μm and an operating speed of 5 μm/s, which can be seen in Supplementary Movie 5. Throughout the above procedure, the fluctuations of normal load were controlled to within ±10 μN. After running-in, the system estimated the friction of flakes under incremental normal loads of 200 μN, 300 μN, 400 μN, 500 μN. For each normal load, the measurement involved 50 reciprocating motions at a speed of 5 μm/s and an amplitude of 5 μm, respectively. The friction force was calculated from the lateral force loop, with some results presented in Fig. 4b. The friction can be modeled as: $f = \mu f_N + f_0$, where $\mu$ represents the differential friction coefficient, $f_N$ stands for the normal load and $f_0$ is the friction force without normal load, contributed by both the edge and in-plane components[35]. To quantify the contribution of these two components, $f_0$ can be described as:

$$f_0 = ax^2 + bx, \tag{1}$$

where $x$ is the length of graphite flake. Based on the above equation, we obtained the fitted $a = 0.009 \pm 0.008$ μN/μm² and $b = 0.173 \pm 0.098$ μN/μm, respectively. The results of the friction $f_0$ and fitting for diverse flake sizes are shown in Fig. 4c. An analysis of the obtained results reveals that the edge friction is about double the in-plane friction, even in the case of a 10 μm flake, indicating the dominant influence of edge friction on $Si_3N_4$ substrate. Furthermore, the differential friction coefficients ($\mu$) of the various-sized graphite flakes are presented in Fig. 4d. It is noteworthy that the probability distribution of the flakes' coefficient doesn't show a significant correlation with flake size. In addition to the friction test, the automated measurements of self-retraction force were conducted with detailed experimental procedures and results presented in Supplementary Note 4 and Supplementary Fig. 10.

Our experimental findings propose that despite the formation of a new interface, the superlubricity state is not completely abolished. Capitalizing on the newly developed auto-friction estimate functions, the system can not only screen out graphite flakes that comply with the conditions for structural superlubricity on fresh interfaces, but also efficiently measure the frictional behavior of SSL materials under higher loads across diverse substrates.

### Demonstrations of the capability of the automated system

The integrated system, i.e., software together with hardware, provides us unprecedented ability to transfer and measure the tribological property of SSL materials with high efficiency and accuracy. For example, the transfer of a large number of SSL flakes is critical in achieving macroscale SSL[1]. However, with manual operation, this process is not only tedious but also sometimes becomes impractical (Supplementary Note 5). With the automated system, however, the challenge can be solved easily. Figure 5a shows a practical demonstration, where within 100 mins, the system transferred 100 SSL

flakes onto etched silicon columns for further connection and scale-up. The efficiency is 15 times faster than the manual operation and with no error. In the realm of large-scale SSL applications, in combination with the bonding technique, an assembly of 25 graphite flakes was achieved based on this transfer system. As shown in Fig. 5b, which depicts the inverted assembled sample under the scrutiny of a 20× microscope, each of the 25 graphite flakes, each 10 μm in size, were meticulously bonded to an Au/Sn alloy. The diagram of assembly and the influence of flake thickness were analyzed in Supplementary Note 6 and Supplementary Fig. 11. In addition to assembly, the precise transfer for SSL materials is also essential for multiple fabrication of SSL devices. Furthermore, the automated system proves itself indispensable in the realm of SSL device fabrication. It ensures a high level of precision in the transfer of SSL materials, an essential process for the successful fabrication of multiple devices. This precision transfer is illustrated in Fig. 5c. Here, the system utilized the designed plate and the coordinates of the graphite flakes within it (left) to automatically calculate the target position for the transfer. To achieve this, the system calibrates the position of the marker in the plate, culminating in the successful and exact transfer of the graphite flake (right) through the image marker calibration. This example provides a testament to the system's intelligent design and precise functionality.

Given the feasibility of scale production for the automated device, it is reasonable to assume that the macroscale SSL system is within reach. Further, with the accumulation of the measurement results for various potential SSL systems, big data can be obtained for SSL. With deep learning, the research and development process of the SSL industry can be greatly promoted.

## Discussion

By implementing the techniques described above, automatic selective transfer and tribological property measurement of various SSL materials have been achieved. Here, we would like to discuss the current limitations of the device as well as the outlook for future improvements. The final alignment accuracy gained with an alignment algorithm becomes less than ±0.5 μm (Fig. 3). However, without human intervention, the angular error may be greater than 5° due to the rotation of the graphite flakes during the pick-up operation. For SSL in heterojunctions, the effect of the interlayer misalignment angle on friction is now found to be much less than for graphite-graphite homojunction, so that when transferred to a non-graphite substrate, SSL applications do not have stringent requirements on the orientation between the surfaces[2]. Considering the applications of SSL with homojunction and the recent burst of fields such as twistronics[36,37], in the future, controlling the angular alignment will be possibly needed, which could be addressed by additional automatic rotatable platforms and angle recognition algorithms. In addition, the electrical properties of SSL[4,38] attracts a lot of attention recently, and automated electrical measurements will also bring great convenience to the application of SSL. Therefore, we will subsequently add programmable electricity meters, combined with automated operation, to expand the electrical measurement capability of the system.

In summary, we have developed a fully automated robotic system for the selective transfer and measurement of SSL materials through precise automated equipment and robust positioning judgment algorithms. The system is principally suitable for a wide range of manipulation and testing on micron scales that can be transferred or experimented with probes. Our developed system enables broader freedom of material design and offers an unprecedented opportunity to explore the full potential of SSL materials. With this equipment, we can reduce human intervention, freeing researchers from repetitive tasks and allowing them to focus on more intellectually creative tasks. As such, it is a fundamental step toward the realization of large-scale SSL applications, where material screening tasks such as genetic

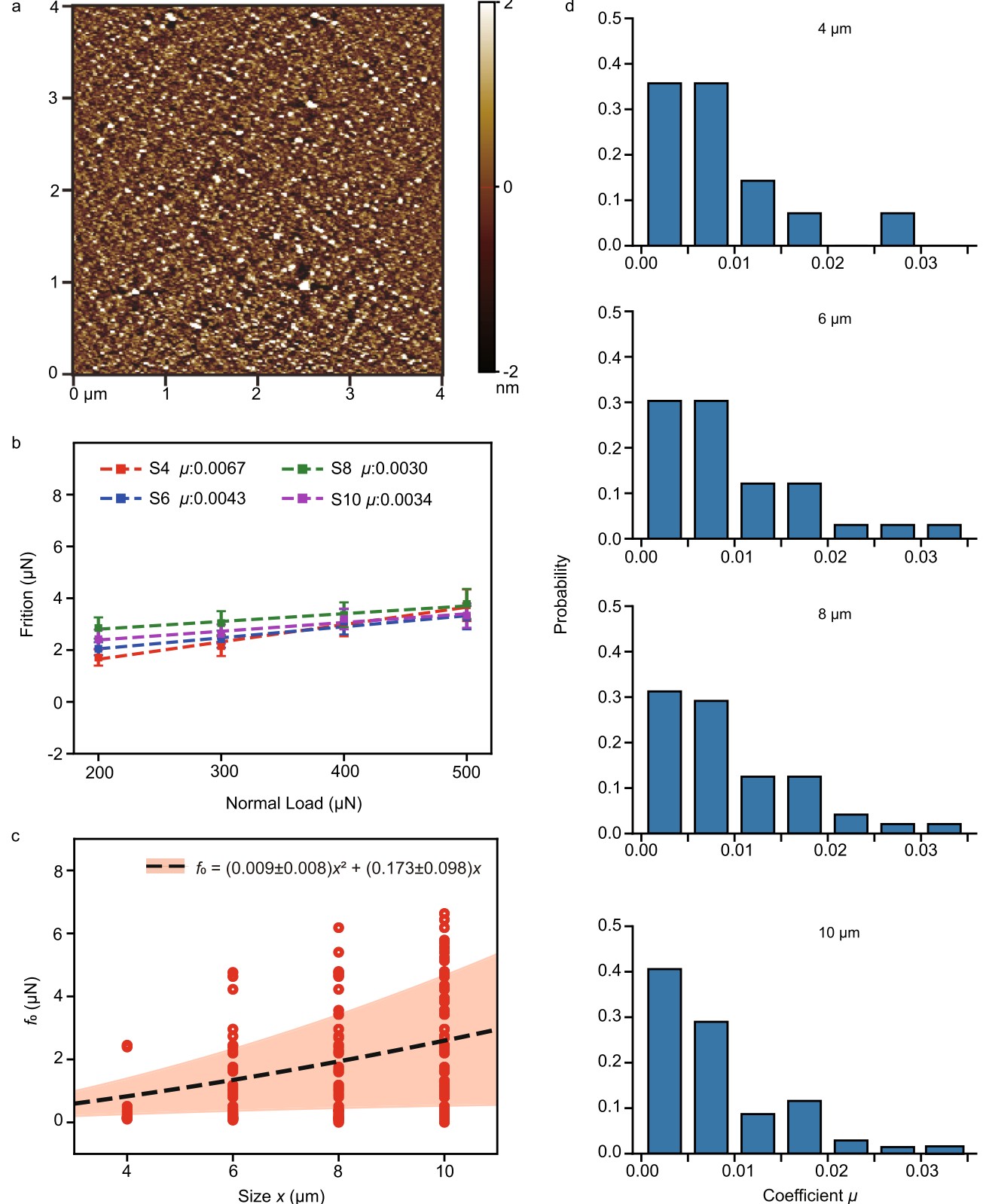

**Fig. 4 | Automated friction measurement for graphite flakes on Si₃N₄ substrate.** **a** AFM height image for the Si₃N₄ surface for $4 \times 4 \ \mu m^2$ range. **b** Friction and the fitted results of some samples with different sizes, where the error bars are calculated as the standard deviation of the 50 independent friction measurements. **c** The friction force without normal load ($f_0$) with the dashed line showing the fitted results and the red shaded band indicating the variance band of the fitted parameters. **d** The friction coefficient ($\mu$) distribution with different flake sizes. Source data are provided as a Source Data file.

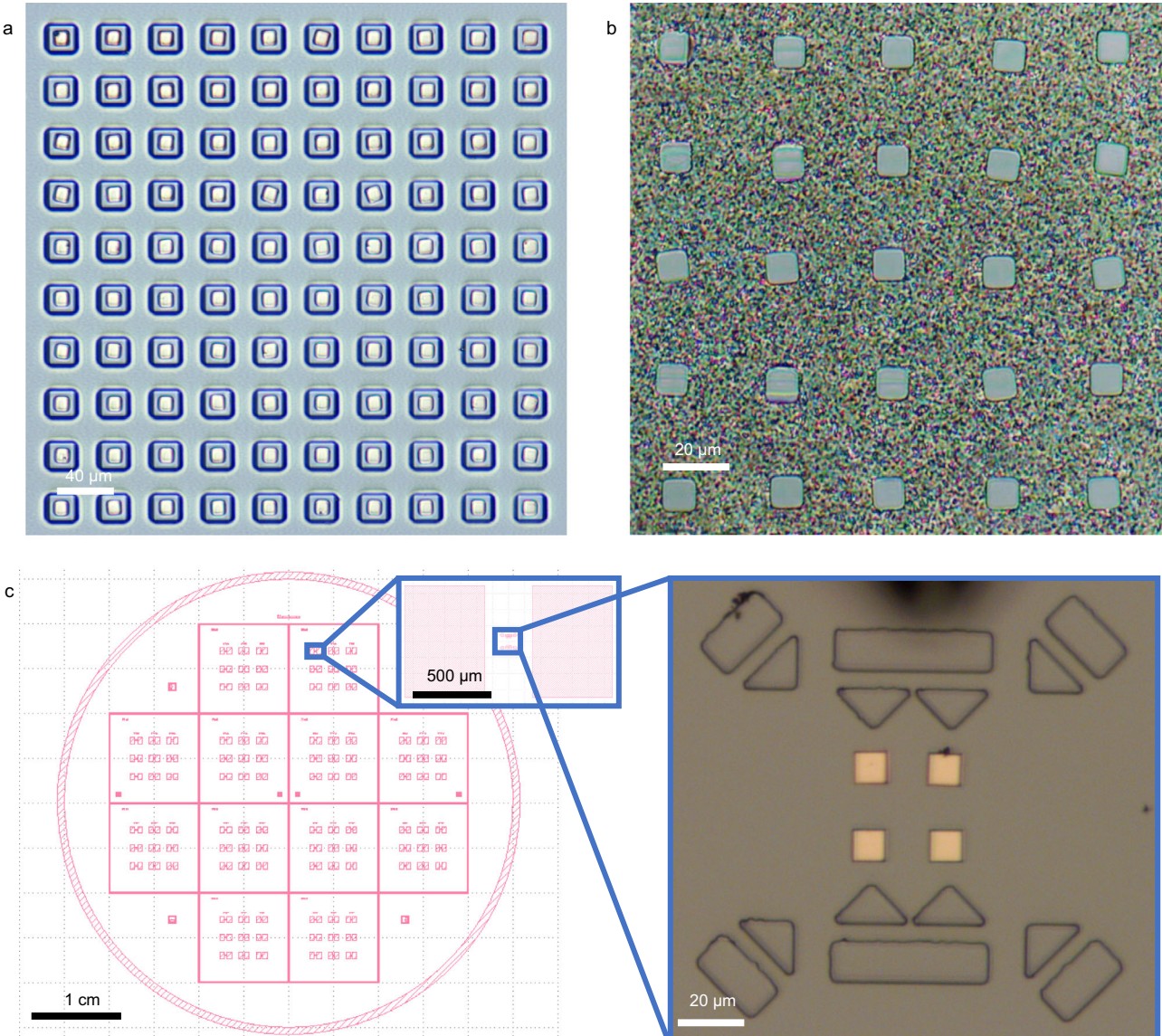

**Fig. 5 | Demonstration for the applications of the automated system. a.** The system transferred 100 SSL flakes onto target silicon columns within 100 mins. **b.** Large-scale assembly of flakes, where 25 flakes were bonded to an Au/Sn alloy. **c.** Wide range of precise transfer of SSL flakes, where flakes were transferred to 2-inch sample according to pre-designed coordination.

engineering of materials or high throughput experiments can be envisaged.

## Methods
### Fabrication of structural superlubric materials
The preparation process flow is shown in Supplementary Fig. 1a. First, highly oriented pyrolytic graphite (HOPG, ZYB grade, Bruker) was dissociated to obtain a clean and flat surface. Then, a layer of Ti or Cr was magnetron sputtered on the freshly cleaved surface of HOPG as an adhesion agent to increase the adhesion between graphite and subsequently deposited films. Then, a 200 nm thick cover layer is grown by magnetron sputtering. The cover layer can be made of conductive materials (such as Au, Ti, Pt, Al, etc.) or insulating materials (such as $SiO_2$, $Si_3N_4$, etc.). The SSL materials with different overlays are shown in Fig. 2c. Next, a photoresist is spin-coated on the surface of the overlay, and an arrayed area of overlay material is obtained by direct laser writing and development. Subsequently, the overlay and adhesion were removed by ion beam etching, and the graphite layer was removed by reactive ion etching with an etching depth of 1 μm. Finally,

the photoresist on the surface is removed to obtain an SSL graphite sheet array with a cover layer. The characterization of the fabricated SSL graphite with Pt film is shown in Supplementary Fig. 1b.

### Hardware and software design
We integrated several key hardware components (an XYZ translation stage, an XYZ precision stage, two force sensors, a microscope objective, etc.) and developed the software system to realize selective transfer and automated experiments for SSL materials. The system's hardware components were connected to a Windows workstation with standard communication ports, including USB, RS-232C and camera link and interacted with software using dynamic link library (DLL) and Serial.

Our customized software system was written in Python and runs based on the software frameworks such as the Nano Position System and the Nanophotonics Lab, which provide dynamic link libraries and core functions for translation stages, enabling modular-based programming and enhancing the robustness and synergy of our system. The GUI, crafted by PyQT5.0 toolkit, enabling the material transfer and tribological property measurement with just a few clicks. The OpenCV-

Python module and Pytorch library[39] were utilized to develop the image processing algorithms and deep learning modules, respectively, with the connections of those algorithms presented in Supplementary Fig. 4.

## Details of hardware in the transfer system and their connections

Figure 1 shows the detailed schematics of our automated transfer system. The core component consists of an *XYZ* translation stage, an *XYZ* precision stage, a CMOS camera, and a tungsten probe with two force sensors (NFS-B, Nators). The range of normal and lateral force sensors are ±50 mN and ±20 mN, respectively and the direction of resolved lateral force is parallel to the tip. The *XYZ* translation stage (Nators, China) for the transfer of SSL material can move simultaneously on the *XYZ* axes with a max speed of 5 mm/s, and the single SSL material can be transferred within 30 s within the range of 50 mm. The *XYZ* precision stage (Piezoconcept, France) for micrometer operations such as graphite flake pick-up has a maximum movement error of 1 nm, enabling precise operations for graphite flakes. To reduce the focusing time, the focus is always 8 μm below the tip during the transfer process. The system records alignment images with ×50 magnification, which are subsequently utilized to calculate the positions of the target flakes relative to the tip.

We provide the information on the hardware connectivity in Supplementary Fig. 5, which shows the detailed composition of hardware and the interaction between hardware and software. After installing the provided software and connecting the hardware, researchers can replicate the functionalities presented in the article.

## Transfer parameters for different types of flakes

Whether to push the graphite flakes and transfer them to the probe depends on the applied normal pressure. Based on this proposal, the correlations between the applied force in pushing, transfer, and the size of graphite flakes have been investigated in Supplementary Fig. 2. We tested many graphite flakes of different sizes and linearly fitted the relationship between maximum normal pressure and size, with the corresponding fitting results shown in the blue lines. In the practical transfer, a 20% margin is preserved for security, which means that the actual parameters during the transfer process were chosen to be calculated according to the orange line.

In addition to the positive pressure setting, the push distance for transferring the graphite flake also needs to be set previously. For judging recovery, the push distance is half the size of the graphite flake and for picking, the push distance should be greater than 0.5 μm of the target material. The speed for both operations is the default 10 μm/s and can be adjusted. The frictions of flakes transferred onto graphite substrates are shown in Supplementary Note 7 and Supplementary Fig. 12, indicating that the transfer process of the system will not affect the SSL performance of the graphite flakes.

## Datasets and settings for training deep networks

We constructed two classification datasets and an object detection dataset for graphite transfer and experiment. For classification datasets, one is for determining whether the material is superlubricity, which contains 571 images of non-superlubricity and 359 images of superlubricity graphite flakes. The other, which judges whether the graphite flake is on the tip, consists of 333 positive sample images with the flake on the tip and 205 negative images. All images in both datasets have a resolution of 140 × 140 and are grayscale. For the object detection dataset, we have collected 60 images with a resolution of 1080 × 1440 containing probe tips and graphite flakes and have labeled each image with the location of the tip and the three types of graphite flakes (good, bad and invalid) with the bounding boxes. Some example images of the datasets are presented in Supplementary Fig. 6.

Based on the above two classification datasets, we trained a superlubricity material selection model and a transfer judgment

model, respectively. For both models, we divided 80% of the data into training sets and trained models with a batch size of 32, the binary cross-entropy loss, and the stochastic gradient descent optimizer with a momentum coefficient of 0.9, with an initial learning rate of 0.01 and cosine annealing for 200 epochs. For the robustness of models, standard data augmentation techniques including random brightness, saturation, hue change, and horizontal flipping were applied for training and central cropping for validation. For material detection, we just followed the default setting in YOLO-V5 except for the batch size of 2. All our models were trained on a 24G NVIDIA RTX 3090 GPU. The loss and accuracy curves at training time are presented in Supplementary Fig. 7 and the final performances in the extra test dataset are presented in Supplementary Note 8 and Supplementary Table 1.

## Automated exceptions handling during transfer assembly

Two different anomalies may occur during the graphite flake transfer assembly, one being contaminants adsorbed on the probe, affecting the van der Waals forces between the graphite flake and the tip, and the other being errors caused by vibration of the translation stages.

To handle the above exceptions, we first designed a self-cleaning module based on electrostatic adsorption. Specifically, once contaminants are found to be adsorbed on the tip, the probe will be moved to a specific copper substrate, and a voltage of 500 V will be applied to adsorb the contaminants through electrostatic interaction. For the second anomaly, the system automatically drops the displacement platform by 10 μm once the pressure recorded by the force sensor exceeds 800 μN and stops the system operation for security.

## Data availability

The data supporting the findings of this study are available within the paper and the Supplementary Information. Source data are provided with this paper.

## Code availability

The software for the automated system is deposited in a DOI-minting repository[40] (Zenodo). The GitHub repository (https://github.com/hsdcl/automated-transfer-and-test-for-SSL) is utilized for subsequent code version management. Custom software with GUI that controls all hardware components is provided with this paper.

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

## Acknowledgements

M.M. acknowledges the financial support from the NSFC (Grant Nos. 11890673 and 51961145304) and the Shenzhen fundamental research key project JCYJ20200109150608043. J.N. acknowledges the financial support from the Shenzhen Science and Technology Program (Grant No. RCBS20210609104540088, JCYJ20210324100600001) and NSFC (Nos. 12204321). Q.Z. wishes to acknowledge the financial support by the National Natural Science Foundation of China Nos. 11572173, 11890671, 51961145304, 11921002. The project is funded by China Postdoctoral Science Foundation (2022T150439)

## Author contributions

L.C., J.N., and M.M. designed the automated system. L.C. constructed all the neural network models and designed the GUI. C.L. and D.S. completed the communication of the various hardware components. J.N. constructed the overall hardware components and completed the data acquisition and analysis. X.H completed the application of precise transfer. L.C. wrote the original manuscript and all authors helped revise it. Q.Z., J.N., and M.M. supervised the research.

## Competing interests

The authors declare no competing interests.
