## [Peer Review File · Nature Communications]

Fully automatic transfer and measurement system for structural superlubric materialsREVIEWER COMMENTS

Reviewer #1 (Remarks to the Author):

The manuscript presents a design of an automated system implying the superlubricity state of layered materials for the fabrication of material patterns. For this, the authors propose selective tungsten-tip-assisted transfer of individual graphite flakes that are picked up from the SSL interfaces. Results demonstrate the ability of the system to create large patterns over up to 4-inch wafers.

As SSL is an interface phenomenon, the whole concept of the study is unclear. The system tries to judge the initial ability of the interface to provide a superlubricity state and then breaks this interface to transfer the top part to the new substrate thus creating the patterns of the flakes. In such case, the new interface is already formed with the substrate material thus eliminating the superlubricity state. I praise the idea of the new applicability of the superlubricity state. However, it seems like the outcomes of such a complicated design approach toward limited material patterning are scant.

My additional concern is that the design lacks control. While the size of the flakes is defined by the initial patterning, the thickness of the transferred flakes is mostly random. Further, from the included images, it seems like the flakes can also undergo some degree of rotation during the transfer.

It is not clear why the judgment of the SSL state is made upon the ability of the flake to return to its original position. As the SSL would enable the initial force-free displacement of the flake, would it be more viable to use this as the confirmation parameter? Meanwhile, the driving force for the flake return could be due to the stretching of bonds formed at edges and defects that facilitate the movement upon release of the tip.

Regarding the applicability of the method, it would be great to include some information on the size of the flakes that can be picked up for patterning. This would be limited by both the possibility to encounter defects that prevent the transfer process as well as by the ability of the tip to pick up the flake.

Also, the size of the tip as well as the position of the tip/flake contact may affect the whole use of the design approach. Additional clarification of this point should be included.

The experimental details mention the 10 nm resolution of the method. Does it mean that the minimum gap with the flakes is 10 nm? How was it measured, mostly relying on imaging capabilities? Considering the rotation possibility, how much will it affect the resulting resolution?

Overall, I feel that the study is interesting but very engineering-specific and might be more appropriate for the journal that focuses on the design of experimental systems and approaches of new fabrication methods.

Reviewer #2 (Remarks to the Author):

The manuscript by Li Chen et al. reports on the development of an experimental platform combining micromanipulation actuators and machine learning for the automated mechanical characterization of mesoscale sliding junctions. The latter consist of graphitic micrometric mesas - prepared by top-down methods - that are already known to support Structural Superlubricity (SSL) when sliding over different surfaces (see refs. 2-5 from the manuscript). This is the case provided that conditions of negligible interfacial contamination and reduced interfacial structural anomalies (unintentionally induced either during mesas fabrication or micromanipulation) are fulfilled. Authors point out that this usually occurs for 60% of the devices prepared by state-of-the-art methods. Given this background, the manuscript shows that contact junctions supporting SSL are

recognized with over 98% accuracy by the deep-learning algorithms from a large population of fabricated devices. Also, the automated method is a factor 15 faster than the same human operations.

The manuscript makes a step forward in the application of SSL devices, by implementing a practical system for the automated transfer and mechanical performance evaluation of contact interfaces. This is very relevant to move SSL from a tailored laboratory set-up to massive real-world applications. The potential of SSL graphitic mesas in tribology, tribogeneration and MEMS technology was recently documented by some of the authors (see refs. 4,5 from the manuscript). In principle, the approach described in this manuscript is suitable to test different types of layered heterointerfaces in a combinatorial fashion, with the goal to speed up the selection of different contact junctions showing SSL. Furthermore, the automated method makes easier the realization of millimeter-scale sliding interfaces based on manipulation of thousands of SSL graphitic mesas – this being a target very difficult to achieve by human operation only.

The main conclusions of the work are well supported by the reported evidences.

The experimental setup is reasonably explained by the authors, both in terms of hardware and software core components. Flow diagrams are provided to describe the most relevant micromanipulation tasks (e.g. see Fig. 2) and more technical details are contained in the Methods section. Section named “Deep learning -based localization , selection, and operation” at pag. 5 comments on the automated image analysis step for judgement purposes (SSL yes/no; picked flake yes/no).

Regarding the methodology, some improvements are possible to the manuscript. Concerning the image analysis steps, the authors should consider to improve understanding of the following issues:

- How do the optical microscope spatial resolution (e.g. objective magnification, illumination conditions etc.), unintentional flakes rotations, unintentional micrometric surface particulates (as those shown in Fig. 2e) and pixel resolution of the acquired images impact the efficiency of the deep learning algorithm?
- How many images are captured for evaluating the sliding state of each flake in SSL? Is the flakes dynamics during tip pushing (or picking) taken into account, or only initial and final flake positions are taken into consideration for decision purposes?

Also, I assume from the manuscript content that the force signals (i.e. both normal and lateral forces) are not used as inputs to train the deep learning algorithm. In other words to effectively push each flake, evaluate SSL and perform the ‘pick & transfer’ operations, normal and lateral forces are not considered as useful inputs for automatic decision making. Only optical images are exploited to this purpose. Regarding this issue:

- Figure 4d shows that forces can be monitored in the GUI. However it is not clear to the Reader how such info is used by the automated algorithm or by the user.
- Avoiding to use force signals for the automated operations seems somehow to limit the potentiality of the fully automatic transfer and measurements system. Can authors comment on this issue? Do they think that using force signals together with optical images might improve the accuracy of the deep-learning algorithms beyond 98% ?

Reviewer #3 (Remarks to the Author):

The authors have demonstrated an automated system which can assemble test samples for structural superlubric materials, and have implemented machine learning protocols to guide its operation. I will refer to this as the Auto SSL system in my review. These results are interesting on a technical level. The area of autonomous fabrication and testing, especially in the context of nanomaterials and interfaces, has substantial promise. However, my impression from the current manuscript is that the results do not yet rise to the full potential that I would expect from such a system, particularly for publication in a high impact journal like Nature Communications. My core concern is as follows: Although the authors can automate the testing of islands for superlubric behavior, and automate the fabrication of complex geometries, I am not sure that these results

will be interesting to a general audience. Furthermore, it is unclear how these demonstrations translate specifically to new scientific results. I feel that the manuscript as it stands is well suited towards an instrumentation-focused journal. However, I am optimistic that the authors may be able to address this concern.

On similar lines, it is unclear if the results can be extended to understand an unknown SSL system. Beyond the demonstrations of automated movement and deep learning augmented characterization, which show that the system can perform the necessary basic operations on well-understood surfaces, it would be useful to see a demonstration of something new, to show that transfer to a heterogeneous surface is functionally viable for SSL experiments.

Additional comments:

Fig. 5b caption should be more specific about the experiment, to save reader trouble of finding the main text paragraph.

Is there any monitoring or control of the relative humidity? Is this important?

What is the resolution of force, range of minimum/max force, and the directions of resolved force?

The authors should provide a simplified diagram of the axes of motion for the various components (i.e., Fig. 1a is nice but too busy, an additional diagram with translation and rotation axes indicated would be helpful). It is otherwise somewhat unclear how many mechanical degrees of freedom exist in the system, which ones are automated, etc.

Lines approx. 60-70: the authors contrast their technique with methods for LED microtransfer printing and 2D materials heterostructure fabrication. However, I think this is a missed opportunity. The introduction text could be re-organized to state why these previous systems are not a good match for the SSL experiments, and what characteristics are specifically needed to define a good system for testing SSL. For example, I am guessing that the nature of the probe interface is important, and neither microLED nor 2D transfer has the appropriate kind of probe. I think one of the most important differences from prior techniques is the need for micronewton force-resolved manipulation. At the same time, these other fields (2D materials particularly) may benefit from the insights of this manipulation technique. It may be useful to draw comparisons with new capabilities that this method brings to the field of automated micromanipulation more generally.

Reviewer #1 (Remarks to the Author):

The manuscript presents a design of an automated system implying the superlubricity state of layered materials for the fabrication of material patterns. For this, the authors propose selective tungsten-tip-assisted transfer of individual graphite flakes that are picked up from the SSL interfaces. Results demonstrate the ability of the system to create large patterns over up to 4-inch wafers. Overall, I feel that the study is interesting but very engineering-specific and might be more appropriate for the journal that focuses on the design of experimental systems and approaches of new fabrication methods.

Response:

We thank the reviewer very much for the assessment of our work. The reviewer's comprehension of our proposed automated system is very precise and insightful. We are also grateful to the reviewers for pointing out the shortcomings in the scientific value of the system. To demonstrate the value of our automated transfer system for SSL research, we have developed the system's capability to measure the tribological properties of SSL materials on the transferred substrate.

Comment 1:

As SSL is an interface phenomenon, the whole concept of the study is unclear. The system tries to judge the initial ability of the interface to provide a superlubricity state and then breaks this interface to transfer the top part to the new substrate thus creating the patterns of the flakes. In such case, the new interface is already formed with the substrate material thus eliminating the superlubricity state. I praise the idea of the new applicability of the superlubricity state. However, it seems like the outcomes of such a complicated design approach toward limited material patterning are scant.

Response:

We thank the reviewer for pointing out our shortage in demonstrating the superlubricity state in the newly formed interface. Admittedly, the tribological property of the transferred flakes on the new

substrate was unknown to the previously designed transfer system, which made the outcomes of the designed pattern formed by the system seem deficient.

Thus, to measure the tribological properties of the newly formed interface with each graphite flake, we have developed the automated friction test function based on the system and tested for more than 100 self-retracted flakes on the transferred Si₃N₄ substrate, with the results shown in Figure R1. The morphology of the transferred Si₃N₄ surface measured by AFM is shown in Fig. R1a, with a corresponding roughness of Ra = 0.73 nm. To measure the friction of the transferred flakes, the system automatically located them and applied a normal load of 200 μN through a tungsten probe. To determine the stable friction force and avoid the interference of factors such as interfacial inclusion, each flake was triggered 200 cycles running-in process¹ on the surface of Si₃N₄ substrate under 200 μN pressure, with an amplitude of 5 μm and an operating speed of 5 μm/s. Throughout the above procedure, the fluctuations of normal load were controlled to within ±10 μN. Then, the system tested the friction of flakes under incremental normal load of 200 μN, 300 μN, 400 μN, and 500 μN. This involved 50 reciprocating motions, with each operating at a speed of 5 μm/s and an amplitude of 5 μm respectively. The friction force was calculated from the lateral force loop, with some results presented in Fig. R1b. The friction can be modeled as: $f = \mu f_N + f_0$, where μ represents the differential friction coefficient, f_N stands for the normal load and f_0 is the friction force without normal load. The differential friction coefficients (μ) of the various sized graphite flakes are presented in Fig. R1d. We found that the probability for the friction coefficient of a flakes below 0.01 is between 60% and 70%. Moreover, no wear was observed across all tribological tests. Our experimental findings propose that despite the formation of a new interface, the superlubricity state is not completely abolished. Capitalizing on the newly developed auto-friction estimate functions, the system can not only screen out graphite flakes that comply with the conditions for structural superlubricity on fresh interfaces, but also efficiently measure the frictional behavior of SSL materials under higher loads across diverse substrates.

Considering the assembled flakes shown in Fig. 5b, the fact that the average friction coefficient of flakes with a given size is below 0.01 shows great significance for achieving macroscale SSL via multi-contact²⁻⁴. Therefore, the ability of automatic selection for SSL graphite and batch transfer of the flakes, together with the assembled technique shown in Fig. 5b, could serve as a viable approach towards macroscale SSL.

Figure R1 is revised Figure 4 in the main article. The above analysis is re-organized and added to the main article on page 2, lines 32-36; page 3, lines 78-82; pages 9-11, lines 187-223.

Figure R1 Automated friction measurement for graphite flakes on Si_3N_4 substrate. **a** AFM height image for the Si_3N_4 surface for $4 \times 4 \mu\text{m}^2$ range. **b** Friction and the fitted results of some samples in different sizes. **c** The friction without normal load (f_0) with the dashed curve showing the fitted results and the red shaded band indicating the variance band of the fitted parameters. **d** The differential friction coefficient (μ) distribution with different flake size.

References:

- 1 Deng, He, et al. "Structural superlubricity in graphite flakes assembled under ambient conditions." *Nanoscale* 10.29 (2018): 14314-14320.
- 2 Li, Panpan, et al. "Toward robust macroscale superlubricity on engineering steel substrate." *Advanced Materials* 32.36 (2020): 2002039.
- 3 Hod, Oded, et al. "Structural superlubricity and ultralow friction across the length scales." *Nature* 563.7732 (2018): 485-492.
- 4 Berman, Diana, et al. "Macroscale superlubricity enabled by graphene nanoscroll formation." *Science* 348.6239 (2015): 1118-1122.

Comment 2:

My additional concern is that the design lacks control. While the size of the flakes is defined by the initial patterning, the thickness of the transferred flakes is mostly random. Further, from the included images, it seems like the flakes can also undergo some degree of rotation during the transfer.

We appreciate the reviewer's concerns about our design's level of control over the flake transfer process. There are several points embedded in this comment, which we would like to address one by one:

1. Flake Thickness: It is correct that the thickness of the transferred flakes is not highly controlled in our current design. However, with bonding technology as shown in Fig. R2a, the variation of the thickness for graphite flakes can be compensated by the deformation of the bonding material, as shown in Fig. R2b. Besides, the randomness of the flake thickness is owing to the brick-stacking structure of highly oriented pyrolytic graphite (HOPG)¹, which could be eliminated with the development of single-crystal graphite fabrication².

The above analysis has been revised and added on page 18 lines 208-219 in Supplementary Information as Supplementary Note 8.

2. Flake Rotation: We admitted that the flakes may undergo some degree of rotation during transfer. This rotation indeed could influence the mechanical properties of flakes on graphite substrate obviously³ due to the interlayer misalignment angle. For our study which primarily focuses on heterojunctions, however, the effects of rotation on friction is not significant⁴. Nevertheless, we acknowledge that control over rotation could be beneficial, especially if our technique were to be

applied to graphite homojunctions in future studies. Therefore, we are planning to incorporate a rotation stage into our design to address this issue. The above analysis has been modified on pages 12-13 and lines 263-269 in the main article.

Figure R2 Large-scale assembly of graphite flakes. **a** Bonding process diagram, the height differences of graphite flakes are resolved by bonding material thickness. **b** Microscope image of graphite flakes array after bonding, where 25 flakes were bonded to an Au/Sn alloy.

References:

- 1 Gongyang, Yujie, et al. "Eliminating delamination of graphite sliding on diamond-like carbon." *Carbon* 132 (2018): 444-450.
- 2 Zhang, Zhibin, et al. "Continuous epitaxy of single-crystal graphite films by isothermal carbon diffusion through nickel." *Nature Nanotechnology* 17.12 (2022): 1258-1264.
- 3 Liu, Ze, et al. "Observation of microscale superlubricity in graphite." *Physical review letters* 108.20 (2012): 205503.
- 4 Song, Yiming, et al. "Robust microscale superlubricity in graphite/hexagonal boron nitride layered heterojunctions." *Nature materials* 17.10 (2018): 894-899.

Comment 3:

It is not clear why the judgment of the SSL state is made upon the ability of the flake to return to its original position. As the SSL would enable the initial force-free displacement of the flake, would it be more viable to use this as the confirmation parameter? Meanwhile, the driving force for the flake return could be due to the stretching of bonds formed at edges and defects that facilitate the movement upon release of the tip.

Response:

We really thank the reviewer for pointing out the potential ambiguity in our method of judging the SSL state. Our rationale for judging the SSL state is based on the ability of the flake to return to its original position, namely the self-retraction motion. As the push distance is on the order of μm which is much larger than the bond length, the driving force for self-retraction motion cannot be the bond stretching. Indeed, previous studies show that the driving force for self-retraction motion is the increase in free energy change due to the newly exposed free surfaces¹. If the friction between the contacting graphite surfaces is smaller than this driving force, the self-retraction motion would occur. In this case, numerous studies show that graphite interface with SSL meet the requirement of small friction^{2,3}. For graphite flake without SSL, such as the presence of external steps, the self-retraction motion will be eliminated³. Therefore, the presence of self-retraction motion could serve as a strong indicator for the presence of SSL.

Due to the free energy of the exposed interfaces, a sufficient lateral force is required for pushing the flake, so that the lateral force pushed by the material is not zero, but a specific value related to the size. Furthermore, if the two interfaces are pushed apart for the first time, the lateral force for the first push is much greater than the self-retraction force due to the energy for breaking the chemical / physical bonds nearing the flake edges⁴, and therefore the initial force-free displacement cannot be a viable parameter for judging the SSL state.

Furthermore, the above analysis is re-organized and added to the main article on page 6 and lines 136-140.

References:

- 1 Wang, Wen, et al. "Measurement of the cleavage energy of graphite." *Nature communications* 6.1 (2015): 7853.
- 2 Liu, Ze, et al. "Observation of microscale superlubricity in graphite." *Physical review letters* 108.20 (2012): 205503.
- 3 Wang, Kunqi, et al. "Characterization of a microscale superlubric graphite interface." *Physical Review Letters* 125.2 (2020): 026101.
- 4 Gongyang, Yujie, et al. "Eliminating delamination of graphite sliding on diamond-like carbon." *Carbon* 132 (2018): 444-450.

Comment 4:

Regarding the applicability of the method, it would be great to include some information on the size of the flakes that can be picked up for patterning. This would be limited by both the possibility

to encounter defects that prevent the transfer process as well as by the ability of the tip to pick up the flake. Also, the size of the tip as well as the position of the tip/flake contact may affect the whole use of the design approach. Additional clarification of this point should be included.

We really appreciate the reviewer for pointing out the lack of analysis on the influence of flake size and tip. Here are our responses to the concerns:

1. Size of flakes: In the main text, we have shown the transfer of graphite flakes with 4-10 μm , which is commonly utilized and has high SSL properties. For graphite flakes smaller than 4 μm , limited by the errors of the localization algorithm, the tip may contact nearing the flake edges when operating, causing a lower transfer rate. For graphite flakes larger than 10 μm , as shown in the Figure R3a, we fabricated some graphite flakes with the size of 20 μm and found that the success rate of picking the graphite flakes is low, probably due to insufficient van der Waals forces between the tip and the mesa. However, previous research indicated that the SSL probability for 20 μm graphite flakes is almost zero, causing low research value¹.

2. Size of the tip: The size of the probe tip has a relatively small impact on the transfer of graphite flakes. However, to improve the success rate of the transfer, probes with different curvature radii should be chosen for graphite flakes of different sizes. When the size of the graphite flake is smaller than 6 μm , it is preferable to use a probe with a curvature radius of 2 μm , as this facilitates precise positioning. When the graphite island is larger than 6 μm , it is preferable to choose a probe with a curvature radius of around 5 μm . This increases the van der Waals force between the graphite flake and the probe tip, thereby increasing the success rate of the transfer.

3. Position of the tip/flake contact: In Figure R3b, we show how to calibrate the tip position. First, the tip is pressed onto a highly oriented pyrolytic graphite (HOPG), creating a depression (left). Then, the tip is lifted and moved 10 μm to the left. The depression location is recorded as the actual contact position and used to train the tip positioning algorithm. Therefore, the predicted tip position corresponds to the contact position between the tip and the flake mesa. During the transfer process, we move the tip to the center of the graphite flake. Considering a positioning error of 1 μm , the contact position of the tip/flake contact has a range of 1 μm around the center of the flake. Due to

the final transfer success rate of over 94%, the 1 μm error has little effect on the transfer success rate.

The above analysis has been reorganized and added in the Supplementary Information on pages 16-18, lines 178-206 and masked in red as Supplementary Note 7.

Figure R3 The influence of flake size and tip location to the transfer process. **a** picking result for 20 μm graphite flake. **b** illustration of calibrating the tip position.

References:

1 Liu, Ze, et al. "Observation of microscale superlubricity in graphite." *Physical review letters* 108.20 (2012): 205503.

Comment 5:

The experimental details mention the 10 nm resolution of the method. Does it mean that the minimum gap with the flakes is 10 nm? How was it measured, mostly relying on imaging capabilities? Considering the rotation possibility, how much will it affect the resulting resolution?

Response:

We appreciate the reviewer's keen observation and we apologize for the confusion caused by the term "10 nm resolution". Upon reviewing the reviewer's comment, we realize the term has been misleading and needs more specific clarification:

1. Resolution of the method: When we mentioned "10 nm resolution", we were referring to the minimum step size that our automated transfer stages can achieve, rather than the feature size that can be patterned or the gap between the flakes. This resolution allows us to position and control the stage (and thus the tip/flake) with very high precision. we are sorry for the ambiguity in our original wording and the context in the main text has been modified on page 4 and lines 95-97.

2. Minimum gap between flakes: we appreciate that the reviewer raised a valuable point regarding the minimum gap between flakes. This gap is largely determined by the localization algorithm we use. Considering the max possible rotation of about 30° , the minimum gap is about 4 μm .

Reviewer #2 (Remarks to the Author):

The manuscript by Li Chen et al. reports on the development of an experimental platform combining micromanipulation actuators and machine learning for the automated mechanical characterization of mesoscale sliding junctions. The latter consist of graphitic micrometric mesas - prepared by top-down methods - that are already known to support Structural Superlubricity (SSL) when sliding over different surfaces (see refs. 2-5 from the manuscript). This is the case provided that conditions of negligible interfacial contamination and reduced interfacial structural anomalies (unintentionally induced either during mesas fabrication or micromanipulation) are fulfilled. Authors point out that this usually occurs for 60% of the devices prepared by state-of-the-art methods. Given this background, the manuscript shows that contact junctions supporting SSL are recognized with over 98% accuracy by the deep-learning algorithms from a large population of fabricated devices. Also, the automated method is a factor 15 faster than the same human operations.

The manuscript makes a step forward in the application of SSL devices, by implementing a practical system for the automated transfer and mechanical performance evaluation of contact interfaces. This is very relevant to move SSL from a tailored laboratory set-up to massive real-world applications. The potential of SSL graphitic mesas in tribology, tribogeneration and MEMS technology was recently documented by some of the authors (see refs. 4,5 from the manuscript). In principle, the approach described in this manuscript is suitable to test different types of layered heterointerfaces in a combinatorial fashion, with the goal to speed up the selection of different contact junctions showing SSL. Furthermore, the automated method makes easier the realization of millimeter-scale sliding interfaces based on manipulation of thousands of SSL graphitic mesas – this being a target very difficult to achieve by human operation only.

The main conclusions of the work are well supported by the reported evidences. The experimental setup is reasonably explained by the authors, both in terms of hardware and software core components. Flow diagrams are provided to describe the most relevant micromanipulation tasks (e.g. see Fig. 2) and more technical details are contained in the Methods section. Section named “Deep learning -based localization, selection, and operation” at page. 5 comments on the automated image analysis step for judgement purposes (SSL yes/no; picked flake yes/no).

Response:

We appreciate the reviewer very much for her/his thorough comments on our work. Her/His understanding of the strategy of our automated system is comprehensive and precise. The reviewer's careful reading and checking will surely help us greatly improve the quality and significance of our manuscript.

Comment 1:

How do the optical microscope spatial resolution (e.g. objective magnification, illumination conditions etc.), unintentional flakes rotations, unintentional micrometric surface particulates (as those shown in Fig. 2e) and pixel resolution of the acquired images impact the efficiency of the deep learning algorithm?

Response:

We appreciate the reviewer's insightful query about the potential factors that may affect the efficiency of our deep learning algorithm. Here we take the localization algorithm as an example and show how each of these factors could impact our process:

1. Optical microscope objective magnification and pixel resolution: The extreme spatial resolution of our system is:

$$d = \frac{0.61\lambda}{N.A.} \approx 0.8 \mu m \quad (R1)$$

Where λ is the average wavelength of the visible spectrum and equals 550 nm and N.A. indicates the numerical aperture. Due to the requirement for operating space, the objective lens with a longer working distance is selected, so the N.A. value equals 0.42 for the 50 \times microscope in our system, leading to the extreme spatial resolution being 0.8 μ m. The spatial resolution of the acquired image, determined by factors such as objective magnification and pixel resolution, plays a crucial role in the deep learning algorithm. For objective magnification, Fig. R4a shows the 8 μ m flakes at 10 \times (left), 20 \times (middle), and 50 \times (right), and the localization algorithm can identify the location of graphite flakes and select those that are not contaminated or damaged. But considering the scale of the image, 50 \times is utilized as the final magnification for localization. And for pixel resolution, Fig. R4b presents the images with different pixel resolutions of 720 \times 960 (left), 1080 \times 1440 (middle), and 1440 \times 1920 (right). The results show that higher pixel resolution contributed to more precise flake bounding boxes.

2. Illumination condition of the microscope: Figure R4c showcases the localization results under varying illumination conditions. Our algorithm demonstrates resilience to changes in lighting conditions, a feature largely attributable to the use of data augmentation methods such as luminance transformation during the training phase. This ensures that the algorithm maintains its performance and continues to deliver accurate results irrespective of the lighting conditions, thereby underlining its robustness and reliability.

The above analysis has been added on the page 14 lines 127-148 in Supplementary Information as Supplementary Note 5 and Fig. R4 has been modified as Supplementary Figure 9

3. Unintentional flake rotations and Unintentional micrometric surface particulates:

Unintentional rotations of the flakes and micrometric surface particulates, such as those shown in Fig. 2e, could indeed affect the performance of our algorithm. To reduce the effect of these graphite flakes on transfer, when positioning graphite flakes, the model classifies them as "invalid" or "bad" graphite flakes when they have significant rotation or contamination, and thus do not perform the transfer. The above analysis has been re-organized and added in the caption of figure 2 in the main text on page 7, lines 159-161.

Figure R4 The influence of optical condition to deep learning algorithm. **a** different objective magnitude. **b** different pixel resolution. **c**. different illumination condition.

Comment 2:

How many images are captured for evaluating the sliding state of each flake in SSL? Is the flakes dynamics during tip pushing (or picking) taken into account, or only initial and final flake positions are taken into consideration for decision purposes?

Response:

The reviewer pointed out the confusion in determining self-retraction, and we are sorry for losing a clear explanation. The algorithm doesn't take additional images in pushing for SSL judgment because the trajectory of flakes during pushing would be restricted by the tip, resulting in no significant difference in the motion of the flake whether it is SSL, as shown in Fig. R5. As for the additional images in self-retraction motion, previous research shows that the velocity of the self-retraction motion of the flake is greater than 1mm/s, causing the entire motion to be completed in approximately 5ms. This duration is too brief for our microscope to capture the full dynamics during the pushing or picking operations. Consequently, we base our evaluations on the initial and final positions of the flakes. These positions provide adequate data to assess the superlubricity state, while also ensuring efficiency in our system operations.

This point has been clarified in the main text on page 6 and lines 139-140.

Figure R5 Microscope images during pushing operation. The results reveal that the trajectory of flake during pushing would be restricted by the tip and consequently, no obvious distinction in the motion of

flake whether it is SSL.

Comment 3:

Also, I assume from the manuscript content that the force signals (i.e. both normal and lateral forces) are not used as inputs to train the deep learning algorithm. In other words, to effectively push each flake, evaluate SSL and perform the ‘pick & transfer’ operations, normal and lateral forces are not considered as useful inputs for automatic decision making. Only optical images are exploited to this purpose. Regarding this issue:

- Figure 4d shows that forces can be monitored in the GUI. However, it is not clear to the Reader how such info is used by the automated algorithm or by the user.

- Avoiding to use force signals for the automated operations seems somehow to limit the potentiality of the fully automatic transfer and measurements system. Can authors comment on this issue? Do they think that using force signals together with optical images might improve the accuracy of the deep-learning algorithms beyond 98%?

Response:

We appreciate for reviewer’s insightful questions regarding the role of force signals in our system. Admittedly, the force signals are not processed as input in deep learning algorithms, but have been utilized for operation and tribological measurement which is incorporated in this revision. There are several points embedded in this comment, which we would like to address as follows:

1. Utilization of force signals: We apologize for not indicating clearly how the users and the algorithm can use the force signal. In the updated tribological test GUI interface (Fig. R6), we have added a “Record” button for saving force signals. In the algorithm, the normal and lateral force signals are sent in real-time to the XYZ precision stage to complete the micro-newton level operations, while the lateral force signals are automatically used to calculate friction, etc. during the tribological measurement. Fig. R6 has been added in the Supplementary Figure 3 and the above analysis has been modified in Supplementary Note 2 on page 11 and lines 93-98.

2. Reason for not using force signals: We appreciate the reviewer's suggestion to improve the accuracy of the algorithm by utilizing lateral force. For self-retraction judgments, it is necessary to push away the graphite flake, in which the breakage of physical/chemical bonding nearing the edge

is required once pushing away for the first time, and previous research shows that the force for breaking bonds is more than one order of magnitude higher than the self-retract force¹, so the lateral force during the pushing process is not a basis for the self-retraction judgment of the graphite flake.

Figure R6 a Additional form for displaying force sensor signals in real-time, where user can save signals through “Record” button. **b** GUI of Automatic tribological test, where selected functions can be operated to multiple flakes.

References:

1 Gongyang, Yujie, et al. "Eliminating delamination of graphite sliding on diamond-like carbon." *Carbon* 132 (2018): 444-450.

Reviewer #3 (Remarks to the Author):

The authors have demonstrated an automated system which can assemble test samples for structural superlubric materials, and have implemented machine learning protocols to guide its operation. I will refer to this as the Auto SSL system in my review. These results are interesting on a technical level. The area of autonomous fabrication and testing, especially in the context of nanomaterials and interfaces, has substantial promise. However, my impression from the current manuscript is that the results do not yet rise to the full potential that I would expect from such a system, particularly for publication in a high impact journal like Nature Communications.

Response:

We are grateful for the reviewer's assessment of our work. The reviewer's comprehension of our proposed automated system is very precise and insightful. We are also grateful to the reviewers for pointing out the shortcomings in rising the full potential of the system in SSL research. To enhance the experimental capability of our automated transfer system, we have developed the system's function to measure the tribological properties of SSL materials on the transferred substrate and completed multiple flakes assembly based on the system.

Comment 1:

My core concern is as follows: Although the authors can automate the testing of islands for superlubric behavior, and automate the fabrication of complex geometries, I am not sure that these results will be interesting to a general audience. Furthermore, it is unclear how these demonstrations translate specifically to new scientific results. I feel that the manuscript as it stands is well suited towards an instrumentation-focused journal. However, I am optimistic that the authors may be able to address this concern. On similar lines, it is unclear if the results can be extended to understand an unknown SSL system. Beyond the demonstrations of automated movement and deep learning augmented characterization, which show that the system can perform the necessary basic operations on well-understood surfaces, it would be useful to see a demonstration of something new, to show that transfer to a heterogeneous surface is functionally viable for SSL experiments.

Response:

We thank the reviewer's concerns about the applicability and interest of our results to a general

audience and their contribution to new scientific results. Regarding the application to an unknown SSL system, we have developed the automated friction test function based on the system and focused on the graphite-Si₃N₄ heterogeneous surface by testing the friction performance for more than 100 self-retracted flakes on Si₃N₄ substrate, with the results shown in Figure R7. For a flat Si₃N₄ surface, with the AFM height image shown in Fig. R1a and roughness of Ra = 0.73 nm, the system automatically measured the friction of each flake. Specifically, the system automatically located the transferred flakes and applied a normal load of 200 μN through a tungsten probe. To determine the stable friction force and avoid the interference of factors such as interfacial inclusion, each flake is triggered 200 cycles running-in process¹ on the Si₃N₄ substrate, with an amplitude of 5 μm and an operating speed of 5 μm/s. Throughout the above procedure, the fluctuations of normal load were controlled to within ±10 μN. Afterwards, the system estimated the friction of flakes under incremental normal loads of 200 μN, 300 μN, 400 μN, and 500 μN. For each normal load, the test involved 50 reciprocating motions, with each operating at a speed of 5 μm/s and an amplitude of 5 μm respectively, where the friction force was calculated from the lateral force loop, with some results presented in Fig. R7b. The friction can be modeled as: $f = \mu f_N + f_0$, where μ represents the differential friction coefficient, f_N stands for the normal load and f_0 is the friction force without normal load, contributed by both the edge and in-plane components. To quantify the contribution of these two components, f_0 can be described as:

$$f_0 = ax^2 + bx, \quad (R2)$$

where x is the length of graphite flake. Based on the above equation, we obtained the fitted $a = 0.009 \pm 0.008 \mu\text{N}/\mu\text{m}^2$ and $b = 0.173 \pm 0.098 \mu\text{N}/\mu\text{m}$ respectively. The results of the friction f_0 and fitting for diverse flake sizes are shown in Fig. R7c. An analysis of the obtained results reveals that the edge friction is about double the in-plane friction, even in the case of a 10 μm flake, indicating the dominant influence of edge friction on Si₃N₄ substrate. Furthermore, the differential friction coefficients (μ) of the various-sized graphite flakes are presented in Fig. R7d. It is noteworthy that the probability of the flakes' coefficient falling below 0.01 ranged between 60%-70% and did not show a significant correlation with flake size.

Figure R7 is new Fig. 4 in the main article. Besides, the above analysis is reorganized and added to the main article on page 2, lines 32-36; page 3, lines 78-82; pages 9-10, lines 187-223.

Figure R7 Automated friction measurement for graphite flakes on Si_3N_4 substrate. **a** AFM height image for the Si_3N_4 surface for $4 \times 4 \mu\text{m}^2$ range. **b** Friction and the fitted results of some samples with different size. **c** The friction without normal load (f_0) with the dashed curve showing the fitted results and the red shade band indicating the variance of fitted parameters. **d** The friction coefficient (μ) distribution with different flake size.

References:

1 Deng, He, et al. "Structural superlubricity in graphite flakes assembled under ambient

conditions." *Nanoscale* 10.29 (2018): 14314-14320.

Comment 2:

Fig. 5b caption should be more specific about the experiment, to save reader trouble of finding the main text paragraph.

Response:

We appreciate the reviewer for pointing out the lack of specificity in Fig. 5b caption. Due to the additional results such as friction measurement of heterogeneous interface and transfer applications such as multi-flakes assembly, the original Figure 5b was modified as Supplementary Figure 10, and the caption was modified on page 16 and lines 173-176 in Supplementary Information.

Comment 3:

Is there any monitoring or control of the relative humidity? Is this important?

Response:

We appreciate the reviewer for pointing out the lack of description of relative humidity control. As the entire system is placed in a clean room of class 1000, the humidity during operation is controlled at $55\% \pm 5\%$. In previous research, the influence of humidity on friction has been studied and the results showed that the friction under ambient conditions and under a nitrogen environment without annealing is comparable and will decrease after annealing¹. The above relative humidity setting has been added in the main text on page 4 and lines 90-92.

References:

1 Song, Yiming, et al. "Robust microscale superlubricity in graphite/hexagonal boron nitride layered heterojunctions." *Nature materials* 17.10 (2018): 894-899.

Comment 4:

What is the resolution of force, range of minimum/max force, and the directions of resolved force?

Response:

We appreciate the reviewer for bringing up these important parameters. The types of force sensors are both NFS-B and Natars. For the normal force sensor, the resolution is 1 μN , the measurement

range is ± 50 mN, and the direction is the Z axis. As for the lateral force sensor, the resolution is 0.2 μ N, the measurement range is ± 20 mN, and the direction is parallel to the tip. The above information is added in the method part of the main text on page 14 and lines 319-322.

Comment 5:

The authors should provide a simplified diagram of the axes of motion for the various components (i.e., Fig. 1a is nice but too busy, an additional diagram with translation and rotation axes indicated would be helpful). It is otherwise somewhat unclear how many mechanical degrees of freedom exist in the system, which ones are automated, etc.

Response:

We appreciate the reviewer's suggestion to clarify the mechanical degrees of freedom in our system. Both stages (the translation stage and the precision stage) are automated and have XYZ degrees of freedom. The XYZ translation stage has a stroke of 4 inches and is utilized for large-range material transfer, and the XYZ precision stage is utilized for precision operation, such as pushing picking and placement, due to its nanometer control accuracy. Figure R8 is the revised Fig. 1a, where the mechanical degrees of stages are denoted and all those components are automated except for the manual telescopic axis. Besides, a modified illustration of Fig. 1a was revised on pages 3-4 and lines of 85-90 in the main context.

Figure R8 Schematic design of the core components of the presented automatic transfer system comprising a XYZ translation stage, a XYZ precision stage, an optical microscope, and et.al, where all components are controlled by the software except for the manual telescopic axis.

Comment 6

Lines approx. 60-70: the authors contrast their technique with methods for LED microtransfer printing and 2D materials heterostructure fabrication. However, I think this is a missed opportunity. The introduction text could be re-organized to state why these previous systems are not a good match for the SSL experiments, and what characteristics are specifically needed to define a good system for testing SSL. For example, I am guessing that the nature of the probe interface is important, and neither microLED nor 2D transfer has the appropriate kind of probe. I think one of the most important differences from prior techniques is the need for micronewton force-resolved manipulation. At the same time, these other fields (2D materials particularly) may benefit from the insights of this manipulation technique. It may be useful to draw comparisons with new capabilities that this method brings to the field of automated micromanipulation more generally.

Response:

We appreciate the reviewer's valuable suggestions and agree that a more detailed comparison with previous methods can better highlight the unique benefits of our technique for SSL experiments.

Both microLED transfer and 2D heterostructure fabrication methods, while impressive in their respective domains, lack certain capabilities critical for SSL testing. For example, as the reviewer guessed, the probe interface is of paramount importance in SSL transfer and testing. The nature of the probe utilized in our system allows for precision manipulation that is not necessarily a priority or characteristic in large scale microLED or 2D transfer techniques.

One of the most significant distinctions of our system is the ability to conduct micronewton force-resolved manipulation. This ability to precisely control and measure forces in the micronewton range provides accuracy and control in manipulating SSL materials, which enables the functions of SSL materials transfer and tribological measurement. Additionally, while our system was designed with SSL testing in mind, the advancements and techniques it employs could very well benefit other fields, such as the repair of dead pixels in microLED and 2D materials manipulation. The precise force-resolved manipulation our system offers could provide novel insights and capabilities in these areas.

The above analysis has been revised and modified on page 2 and lines 60-72.

REVIEWERS' COMMENTS

Reviewer #1 (Remarks to the Author):

The authors did great job addressing the comments and performing additional needed experiments to clarify the questions raised by the reviewers. My only additional suggestion is to make sure that all the additional information in the response is included in the supplementary materials.

Reviewer #2 (Remarks to the Author):

The revised manuscript and revised Supplementary Information file have satisfyingly addressed my original concerns. I suggest publication of the revised manuscript.

Reviewer #3 (Remarks to the Author):

I appreciate the revisions, which have addressed my concerns. I believe this manuscript is suitable for publication.

Reviewer #1 (Remarks to the Author):

The authors did great job addressing the comments and performing additional needed experiments to clarify the questions raised by the reviewers. My only additional suggestion is to make sure that all the additional information in the response is included in the supplementary materials.

Response:

We appreciate the reviewer's suggestions to include all additional information from our responses in the supplementary materials and have taken steps to ensure that all pertinent details discussed in our response to the reviewers are comprehensively covered in the revised supplementary materials.

Reviewer #2 (Remarks to the Author):

The revised manuscript and revised Supplementary Information file have satisfyingly addressed my original concerns. I suggest publication of the revised manuscript.

Response:

We thank the reviewer for the positive comments on the revised manuscript and Supplementary Information.

Reviewer #3 (Remarks to the Author):

I appreciate the revisions, which have addressed my concerns. I believe this manuscript is suitable for publication.

Response:

We thank the reviewer's kind words and appreciate for confirming that the revised manuscript has successfully addressed the reviewer's concerns.